# Influence of the Immune Microenvironment Provided by Implanted Biomaterials on the Biological Properties of Masquelet-Induced Membranes in Rats: Metakaolin as an Alternative Spacer

**DOI:** 10.3390/biomedicines10123017

**Published:** 2022-11-23

**Authors:** Marjorie Durand, Myriam Oger, Krisztina Nikovics, Julien Venant, Anne-Cecile Guillope, Eugénie Jouve, Laure Barbier, Laurent Bégot, Florence Poirier, Catherine Rousseau, Olivier Pitois, Laurent Mathieu, Anne-Laure Favier, Didier Lutomski, Jean-Marc Collombet

**Affiliations:** 1Osteo-Articulary Biotherapy Unit, Department of Medical and Surgical Assistance to the Armed Forces, French Armed Forces Biomedical Research Institute, 91223 Brétigny-sur-Orge, France; 2Imaging Unit, Department of Platforms and Technology Research, French Armed Forces Biomedical Research Institute, 91223 Brétigny-sur-Orge, France; 3Tissue Engineering Research Unit-URIT, Sorbonne Paris Nord University, 93000 Bobigny, France; 4Molecular Biology Unit, Department of Platforms and Technology Research, French Armed Forces Biomedical Research Institute, 91223 Brétigny-sur-Orge, France; 5Laboratoire Navier, Gustave Eiffel University, Ecole des Ponts ParisTech, CNRS, 77447 Marne-la-Vallée, France; 6Department of Surgery, Ecole du Val-de-Grace, French Military Health Service Academy, 1 Place Alphonse Laveran, 75005 Paris, France

**Keywords:** Masquelet-induced membrane, macrophages, PMMA, metakaolin

## Abstract

Macrophages play a key role in the inflammatory phase of wound repair and foreign body reactions—two important processes in the Masquelet-induced membrane technique for extremity reconstruction. The macrophage response depends largely on the nature of the biomaterials implanted. However, little is known about the influence of the macrophage microenvironment on the osteogenic properties of the induced membrane or subsequent bone regeneration. We used metakaolin, an immunogenic material, as an alternative spacer to standard polymethylmethacrylate (PMMA) in a Masquelet model in rats. Four weeks after implantation, the PMMA- and metakaolin-induced membranes were harvested, and their osteogenic properties and macrophage microenvironments were investigated by histology, immunohistochemistry, mass spectroscopy and gene expression analysis. The metakaolin spacer induced membranes with higher levels of two potent pro-osteogenic factors, transforming growth factor-β (TGF-β) and bone morphogenic protein-2 (BMP-2). These alternative membranes thus had greater osteogenic activity, which was accompanied by a significant expansion of the total macrophage population, including both the M1-like and M2-like subtypes. Microcomputed tomographic analysis showed that metakaolin-induced membranes supported bone regeneration more effectively than PMMA-induced membranes through better callus properties (+58%), although this difference was not significant. This study provides the first evidence of the influence of the immune microenvironment on the osteogenic properties of the induced membranes.

## 1. Introduction

In the face of large bone defects, surgery is required to restore the shape and function of the bone. The induced membrane technique (IMT), also known as the Masquelet technique, is a widely used two-stage surgical procedure. This technique is unique in preparing the bed graft by molding a polymethylmethacrylate (PMMA) spacer to fill the bone defect [1]. The implantation of this spacer leads to the formation of an induced membrane (IM)—granulation tissue surrounding the spacer. In the second step, the spacer is removed while preserving the integrity of the IM. A standard autologous bone graft is then implanted into the IM cavity to repair the bone.

The IM, which acts as a biologically privileged membrane at the site of the defect, is the key element in this procedure. Preclinical and clinical studies have highlighted the various roles of the IM. It prevents graft resorption and muscle invasion of the defective bone by acting as a barrier membrane. It also creates an osteogenic and osteoinductive environment by secreting many growth factors and cytokines. These factors include bone morphogenic protein-2 (BMP-2), interleukin-6 (IL-6), transforming growth factor-β (TGF-β), vascular endothelial growth factor A (VEGF-A), von Willebrand factor (vWF) and metalloproteinase-9 (MMP-9) [2,3,4,5,6]. The IM has also been shown to be highly vascularized and to serve as a source of bone progenitor cells. 

Biologically, the IM results from a foreign-body reaction (FBR). Upon implantation, all biomaterials elicit a FBR, a natural immunoinflammatory process that isolates the implant from the rest of the body in a collagenous capsule [7]. Macrophages are plastic cells that play a key role in the FBR by orchestrating the inflammatory environment around the implanted biomaterial. Indeed, macrophages can adopt diverse functional phenotypes upon activation, ranging from M1 (pro-inflammatory) to the M2 (pro-healing) profiles. Interestingly, macrophage activation depends on the shape and surface properties of the biomaterial (chemistry, porosity, wettability, roughness and stiffness) [8]. 

We hypothesized that changing the chemical composition of the spacer generating the IM in the Masquelet technique would modify the immune microenvironment in which the FBR occurred, thereby altering the osteogenic properties of the IM and potentially enhancing bone regeneration. We tested this hypothesis by replacing the PMMA of standard spacers with an alternative biomaterial, a metakaolin-based geopolymer (Davidovits) [9]. This polymer is synthesized by an alkaline activator solution’s reaction (geopolymerization) with metakaolin particles. Chemically, metakaolin is a dehydroxylated form of the clay mineral kaolinite, an aluminosilicate material. Aluminosilicates and their derivatives are known to have immunostimulatory effects due to induction of macrophage activation [10]. Metakaolin was, therefore, chosen for this study based on its immunogenicity, and its innocuity relative to other clay minerals, such as bentonite [11]. Wiemann et al. [12] recently showed that the intratracheal instillation of kaolin in rats induced transient macrophage-based hypercellularity in rat lungs, with no signs of inflammation or structural change in the lung parenchyma, whereas bentonite instillation leads to a very intense lung inflammation with changes to the structure of the lung epithelium. Metakaolin is listed in the US Pharmacopeia, suggesting that its transfer into clinical practice might be facilitated in terms of the requirements for medical device regulation. Kaolinite and its chemical derivatives have been widely used in the pharmaceutical domain for decades as well-characterized pharmaceutical excipients: diluents, binders, disintegrants, pelleting agents, granulating agents, amorphizing agents, film-coating additives or even drug carriers [13,14]. They are also used as active pharmaceutical ingredients in hemostatic wound dressings, dermatological protectors, gastrointestinal protectors and antidiarrheal agents. 

We used a validated Masquelet model in rats to assess the osteogenic properties of metakaolin-generated IM with histological and immunohistochemical methods. We first analyzed the distribution of M1-like and M2-like macrophage populations within the IM. Finally, we determined the impact of metakaolin spacers as an alternative to PMMA on bone-healing outcomes.

## 2. Materials and Methods

### 2.1. Animals

Animal procedures were approved by the appropriate institutional animal care and use committee (protocol 65 DEF_IGSSA_SP). Interventions were performed at an accredited animal facility. Male Sprague Dawley rats (Charles River, France) were housed individually in cages with controlled temperature and lighting conditions, and food and water supplied ad libitum. The rats were eight weeks old (mean weight of 200 g) when they underwent the first surgical procedure. In cases of postoperative complications, such as deep infection or bone fixation failure, the animals were excluded from the study and euthanized. Animals were killed by the intraperitoneal injection of sodium pentobarbital (150 mg/kg) at the age of 12 weeks (for IM analysis) or 22 weeks (for bone repair assessment). 

### 2.2. Surgical Procedures

IMT surgery (steps 1 and 2) was performed as previously described [15]. The first stage of surgery was performed under general anesthesia induced by the intraperitoneal administration of a ketamine/medetomidine mixture (60 and 0.42 mg/kg, respectively). Rats were placed in the prone position and an incision was made through the skin and muscle to expose the right femur. A mini external fixator (RatExFix RISystem, Davos, Switzerland) was screwed onto the anterolateral surface of the femur shaft, and a Gigli wire saw was used to create a 6 mm segmental defect. The bone defect was filled with either hand-made PMMA or metakaolin spacers (*n* = 5/group). Four weeks later, the animals underwent the second stage of graft surgery or were killed for stage 1 membrane studies. This time point was chosen based on the results of our previous model validation study [15]. For stage 2 of the IMT surgery, rats (*n* = 5/group) were anesthetized with isoflurane (1.5 to 2% isoflurane in 1 to 1.5 L of O_2_/min). An incision was carefully made in the membrane for spacer removal. The defect was then filled with a morselized corticocancellous allograft harvested from the distal femur of littermates killed on the same day. On three consecutive days after each surgical procedure, the animals received subcutaneous injections of a cephalosporin antibiotic (10 mg/kg enrofloxacin) and an opioid painkiller (0.05 mg/kg buprenorphine, twice daily). Unprotected weight-bearing activity was allowed immediately after surgery. The animals were weighed daily, and animal facility staff also evaluated their behavior, pain, normal movements and the appearance of the wound every day. Radiographic follow-up evaluations were performed every two weeks to check for incorrect spacer positioning and implantation failure. The animals were killed after 10 weeks for bone-healing assessment.

### 2.3. Spacers

PMMA (Palacos R + G, Heraeus, Hanau, Germany) spacers were made by hand under sterile conditions before surgery. They were macroscopically smooth and cylindrical. Metakaolin spacers were prepared in advance, as follows. Activated metakaolin paste was prepared by mixing sodium silicate (activating solution) with metakaolin particles. The activating solution was prepared by mixing NaOH solution (mass concentration Cw = 0.35) with a solution containing Na_2_O (Cw = 0.08) and SiO_2_ (Cw = 0.27) provided by MERCK KGaA, and water. The metakaolin particles (Argical M 1200S) were provided by AGS Minéraux (Clérac, France). The Brunauer, Emmett and Teller (BET) specific surface area of these particles was 19 m^2^/g and their mass mean diameter was about 2 µm. The chemical composition of the resulting paste was characterized by the following ratios: Si/Al = 1.71 (molar), Na_2_O/Al_2_O_3_ = 1.01 (molar) and solid/liquid = 1.50 (mass ratio). The paste was poured into cylindrical PMMA molds, all of the same diameter (4 mm), but with three different lengths: 5.5, 6 and 6.5 mm. The opening was covered and the molds were left at room temperature for 48 h. Geopolymerization resulted in very slight shrinkage, facilitating the removal of the metakaolin spacers from the molds. Prior to animal implantation, spacers were exposed to steam sterilization accomplished in an autoclave (20 min, +121 °C).

### 2.4. Bone Turnover Assessment

Serum samples were used to assess bone turnover markers, both markers of formation (procollagen-1 N terminal telopeptide or P1NP) synthesized by osteoblasts, and markers of resorption (tartrate-resistant alkaline phosphatase C or TRAP-C) released by osteoclasts during bone matrix remodeling. The levels of these markers were determined by ELISA. For P1NP assessments, we used the Rat/Mouse PINP EIA^TM^ assay kit (ref. AC-33F1, IDS Inc., El Segundo, CA, USA). The rat TRAPTM (TRAcP-5b) ELISA kit (ref SBTR102, IDS Inc.) was used for TRAP-C assays. Duplicate determinations were performed for each sample, and the two results were then averaged. The P1NP/TRAP-C ratio was calculated to express bone turnover four weeks after creating bone defects.

### 2.5. Histology

Membrane fragments were fixed in a 4% paraformaldehyde solution for embedding in paraffin. Sections (5 µm) were cut and stained with hematoxylin-eosin-saffron (HES) or prepared for BMP-2 immunostaining and CD68 CD206 immunofluorescence analysis. A pathological histologist examined all HES-stained sections. All immuno-stained sections on glass slides were digitized with a Nanozoomer S60 slide scanner (Hamamatsu) to quantify whole-slide images. Scanning resolution at 20× magnification was 0.46 µm/px. Virtual slide images were saved in 16-bit raw format for immunofluorescence analysis and RGB TIFF format for sections with standard staining. All image processing was performed with Fiji software [16]. To quantify IM cellularity, sections were stained with DAPI to visualize the cell nuclei. A region of interest (ROI) was drawn manually to exclude muscle fibers from the areas analyzed. The DAPI image was thresholded with the Triangle algorithm to select the brightest objects. Each object was then isolated to segment clusters of nuclei based on the local maxima of the initial image (with the segmented particles option). A “logical and” was used between the first threshold and the segmented particles.

### 2.6. BMP-2 Immunostaining

We assessed the expression of BMP-2, a potent osteogenic growth factor, by performing immunohistochemical analyses on paraffin-embedded IM sections, as previously described [15]. The rabbit polyclonal antibody specific for BMP-2 (Bioworld 90141) was used at a dilution of 1:200. The ready-to-use ImmPRESS HRP Anti-Rabbit IgG detection kit (Vector, MP-7451) was incubated with the slides for 30 min, and hematoxylin counterstaining was then performed.

### 2.7. Real-Time PCR Analysis

Membrane tissues for molecular biology analysis were collected and stored in RNA later^®^ (Ambion, Austin, TX, USA). Samples were kept at +4 °C for 24 h and then stored at –20 °C until homogenization in guanidium-based lysis buffer with a TissueLyser II (RLT buffer, Qiagen, 20 Hz, 2 min, two 3 mm-carbide beads). According to the manufacturer’s recommendations, total RNA was extracted with the Nucleospin RNA XS kit (Macherey Nagel, France) but with an additional proteinase K digestion step (Qiagen, Les Ulis, France). RNA was eluted in 15 µL of RNase-free water. The quantity and purity of the RNA were determined with an Agilent TapeStation 4200 automated electrophoresis system, with RNA screen tape and reagents (Agilent Technologies, Santa Clara, CA, USA), according to the manufacturer’s instructions. The total RNA concentration of each sample was expressed in nanograms per microliter. RNA quality was assessed by determining the RNA integrity number (RIN) on a scale of 1 (completely degraded RNA) to 10 (intact RNA), as described by Schroeder and collaborators [17]. The mean RIN value was 7.4 for the PMMA group and 7.6 for the metakaolin group. A real-time PCR study was carried out as described in the MIQE guidelines [18].

Based on the manufacturer’s instructions, the first-strand cDNA was generated by reverse transcription with the EuroScript reverse transcriptase on 400 ng total RNA (Eurogentec #RT-RTCK-03, Seraing, Belgium). RNA integrity and reverse transcription yields were confirmed with the 5′/3′ integrity assay and *Rplp0* selected primers (Appendix A) [19]. Primers were designed and optimized with MacVector^®^ 3.5 software (Accelrys, San Diego, CA, USA) to prevent dimerization, self-priming and melting temperature. Primers binding to flanking introns were selected to exclude genomic DNA amplification and were assessed for specificity to rats with the Blast nucleotide algorithm. Oligonucleotide primers were synthesized by Eurogentec (Sereing, Belgium). Real-time qPCR was performed with a LightCycler^®^ 480 instrument (Roche Applied Science, Mannheim, Germany) with SybrGreen I Mastermix (Roche Applied Science). Quantification of mRNA was measured using the comparative threshold method [20] with efficiency correction estimated from a standard curve. The qPCR primers used for the three reference genes (ribosomal protein lateral stalk subunit P0 (*Rplp0*), peptidylprolyl isomerase A (*Ppia*), hypoxanthine phosphoribosyltransferase 1 (*Hprt*)) and the five target genes (transforming growth factor beta 2 (*TGFβ2*), interleukin-6 (*IL-6*), interleukin-1-beta (*IL-1β*), insulin-like growth factor (*IGF1*) and vascular endothelial growth factor A (*VEGF*-*A*) are listed in Appendix A, along with the optimized concentration and annealing temperature for each primer. Normalization was assessed with geNorm software. A geometric mean for the three internally validated reference genes (*Rplp0*, *Ppia* and *Hprt*) was calculated [21]. The pairwise variation of these three genes was 0.119, which is below the threshold (0.15), requiring the inclusion of an additional normalization gene.

### 2.8. Immunofluorescence Assays and Macrophage Quantification

Immunofluorescence analysis was performed to study the phenotypic profiles of the macrophages in the IM. “M1-like macrophages” were defined as CD68-positive cells, whereas “M2-like macrophages” were defined as cells positive for both CD68 and CD206, as previously described [22]. Cells negative for CD68 but positive for CD206 were defined as muscle satellite cells [22,23].

Sections were permeabilized by incubation for 15 min with 0.5% Triton X100 (*v*/*v*) buffered with PBS. Non-specific binding sites were blocked by incubation with Emerald Antibody Diluent (Sigma 936B-08) for 1 h. The sections were then incubated overnight at +4 °C with the primary mouse anti-CD68 (BIO-RAD MCA341GA, Hercules, CA, USA) antibody at a dilution of 1:100 and the primary rabbit anti-CD206 (Sigma HPA045134) antibody at a dilution of 1:100. They were washed in PBS and incubated with an anti-rabbit green fluorescent Alexa Fluor 488 (A-21206, Thermo Fisher Scientific) secondary antibody and an anti-mouse red fluorescent Alexa Fluor 568 (A10037, Thermo Fisher Scientific) secondary antibody, both at a dilution of 1:1000, for two hours at room temperature. Finally, sections were washed in PBS for 20 min and mounted in Fluoroshield mounting medium with DAPI (Abcam, Cambridge, UK, ab104139). Fluorescence was detected under an epifluorescence microscope DM6000 (Leica, Wetzlar, Germany) equipped with monochrome and color digital cameras. Macrophages were quantified on whole-slide images with FIJI software. The M2-like cells displayed double labeling (green + red), whereas M1-like macrophages displayed only red labeling. A “zone of influence” was defined around each nucleus, with nuclei segmented for cellularity measurement as seeds. On Alexa Fluor 488-labeled images, the Otsu method set a double threshold for the previously drawn ROI. On the Alexa Fluor 568-labeled images, an Otsu threshold was determined within the same ROI. A geodesic reconstruction of the cells was performed with each type of immunofluorescence labeling used as a seed and the “zone of influence” of the nuclei as a mask. These analyses yielded the number of stained cells/total number of cells expressed as a percentage. 

### 2.9. Liquid Chromatography–Tandem Mass Spectrometry (LC-MS/MS)

Proteins secreted by IM fragments were identified by mass spectrometry. No labeling/tagging techniques were used in our LC-MS/MS study. Therefore, we could not determine the abundance of the secreted proteins. Instead, we aimed to identify all the secreted proteins and their molecular networks and compare protein secretion frequencies between the two batches. The proteins secreted by IMs were purified by an organic solvent-based protein precipitation method. Briefly, nine volumes of ice-cold acetone-methanol (8:1) were added to one sample volume, and the resulting mixture was incubated overnight at −20 °C. The samples were then centrifuged at 10,000× *g* for 30 min, and the protein pellet was dissolved in 40 µL of 2X Laemmli buffer (Biorad). 

Protein samples were briefly subjected to SDS-PAGE (8% acrylamide gel, 8 × 8 cm) until the sample had completely penetrated the gel. Following in-gel fixation (ethanol 30% *v*/*v*, acetic acid 7% *v*/*v*) for 1 h and protein staining with Coomassie Brilliant Blue, each band was excised manually and cut into small pieces with a scalpel. Gel pieces were dehydrated by incubation in 100 µL acetonitrile for 15 min and rehydrated by incubation with 100 µL 25 mM NH_4_HCO_3_ for 10 min. This operation was repeated twice. After final dehydration in 100 µL acetonitrile, gel pieces were covered with 100 µL 10 mM DTT in 25 mM NH_4_HCO_3_ and incubated at +56 °C for 45 min. The supernatant was removed, and 100 µL of 55 mM iodoacetamide in 25 mM NH_4_HCO_3_ was added. The mixture was left in the dark at room temperature for 30 min and the supernatant was then removed. The gel pieces were covered with 100 µL 25 mM NH_4_HCO_3_ for 10 min and dehydrated by incubation with 100 µL acetonitrile for 15 min. The volume of the dehydrated gel was evaluated and three volumes of trypsin (12 ng/µL) in 25 mM NH_4_HCO_3_ (freshly diluted) were added. The digestion was allowed to proceed at +35 °C overnight. Peptides were finally extracted from the gel pieces by incubation in 60% acetonitrile/5% HCOOH for 1 h. The supernatant was collected, the volume of each peptide sample was reduced to 15 µL, and the peptides were analyzed by mass spectrometry.

Peptide samples were then analyzed with a QToF instrument (Xevo G2-XS QTof, Waters, Milford, MA, USA) coupled to a nano liquid chromatography apparatus (ACQUITY UPLC M-Class system, Waters) running with two buffers: 0.1% formic acid in water (A) and 0.1% formic acid in acetonitrile (B). We separated 3 µL of each sample on a C18 reverse-phase column (NanoE MZ HSS C18 T3, 1.7 μ 75 µm × 100 mm, Waters), with a linear gradient of 5% to 85% buffer B over 120 min at a flow rate of 300 nL min^−1^. Peptide ions were analyzed with Masslynx v4.1, with the following data-independent acquisition steps (DIA): MS scan range: 50–2000 *m*/*z*, scan time 0.5 s, ramp collision energy from 15 to 40 V. Proteins were identified with Progenesis QI for proteomics v3.0 (Waters) with the following parameters: enzymatic cleavage by trypsin with two missed cleavages allowed, carbamidomethylation for cysteine residues and potential oxidation for methionine residues. Only peptides with a score of at least 5 were considered. The Uniprot KB database (www.expasy.org (accessed on 1 October 2019)) and a custom-built contaminant database (trypsin, keratin, etc.) were used. The species of origin was restricted to the rat. The identified proteins were filtered to retain only those with a minimum of three fragments per peptide and one peptide per protein. Analysis was performed on *n* = 5 animals/group. A protein was considered differentially secreted if its detection frequency in a group differed from that of the other group by at least two animals.

### 2.10. MicroCT

Three-dimensional microcomputed tomography (µCT) was used to quantify bone regeneration 10 weeks after stage 2 of the Masquelet technique. The rats were killed, and the limb on which surgery was performed was collected, together with the surrounding soft tissues, and fixed by incubation in 10% phosphate-buffered formalin for two weeks. The area between the inner pins was scanned by microCT (Skyscan 1174, Bruker Micro-CT, Billerica, MA, USA) with a voltage source of 50 keV, a current of 745 mA and an isotropic resolution of 14.4 µm. Three-dimensional reconstruction was performed for all scans and analyzed with the same parameter setup (NRecon v.1.6 and CTAn v.1.11 software, SkyScan, Kontich, Belgium) to separate mineralized elements from the background, with the software histogram tool used to determine grayscale level threshold values. As a dedicated external fixator with a guide saw had been used to create the defect, it was possible to locate the 6 mm long defective region with precision (the distance between the two adjacent pins) and to identify it as the region of interest. The following data were collected within the region of interest: total defect volume (TV in mm^3^) and bone volume (BV in mm^3^) for the calculation of the bone volume ratio (BV/TV, as a %).

### 2.11. Statistical Analysis

All results are reported as means ± standard error of the mean (SEM). The Shapiro-Wilk test was used to determine whether the data followed a normal distribution. An *F*-test was performed to verify the assumption of equal variances. Two-tailed Student’s *t*-tests were used for comparisons if the data met both these requirements (normal distribution and equal variances). If one or both the assumptions were not met, the PMMA and metakaolin groups were compared in non-parametric Mann–Whitney *U*-tests. Values of *p* < 0.05 were considered significant in all tests. Statistical analyses were performed with GraphPad Prism 5 statistical software (GraphPad Software Inc, La Jolla, CA, USA).

## 3. Results

### 3.1. Animals and Blood Parameters at the End of IMT Stage 1

All rats tolerated surgical procedures well and gained weight steadily from day 4 after stage 1 surgery onwards. Two animals (one PMMA and one metakaolin) were excluded from the analysis due to infection-related fixator failure. Given the inflammatory potential of the aluminosilicate present in metakaolin, we determined blood cell counts for the animals to assess systemic inflammation at the time of death. White blood cell counts and red blood cell parameters were similar between the PMMA and metakaolin groups (Figure 1A). Serum P1NP and TRAP-C concentrations and ratios were similar in the two groups, suggesting that bone remodeling activity four weeks after the creation of the bone defect was similar in the PMMA and metakaolin groups (Figure 1B).

### 3.2. Comparison of Biological Properties between Metakaolin- and PMMA-Induced Membranes

#### 3.2.1. Membrane Architecture and Cellularity

We previously showed that bioactive IMs are organized as bilayered structures and have a rich cellular network. Figure 2 illustrates typical sections of PMMA-induced (Figure 2A) and metakaolin-induced (Figure 2B) membranes, with an inner layer in contact with the biomaterial, including fibroblasts, lymphocytes and macrophages. A thick outer layer principally consists of fibroblasts with a dense vascular network in contact with the muscle. The quantification of DAPI-stained nuclei showed cell density to be slightly higher in metakaolin-IMs than in PMMA-IMs, although this difference was not statistically significant (4553 ± 51 nuclei/mm^2^ in the PMMA group versus 5882 ± 695 nuclei/mm^2^ in the metakaolin group, *p* = 0.15).

#### 3.2.2. Gene Expression within Membranes

We compared the expression of key inflammation-related genes involved in wound healing between PMMA-IMs and metakaolin-IMs. Real-time RT-PCR analysis (Figure 3) showed that the relative levels of insulin-like growth factor-1 (*IGF-1*), vascular endothelial growth factor (*VEGF*), interleukin-6 (*IL-6*) and interleukin-1-beta (*IL-1β*) expression was similar in PMMA-IMs and metakaolin-IMs. However, in metakaolin-IMs, transforming growth factor-β (*TGF-β*) mRNA levels were significantly upregulated (fold change = 2.74, *p* = 0.016), potentially enhancing bone healing and regeneration. 

#### 3.2.3. Secretion of Proteins by the IM and BMP-2 Expression within Membranes

IMs form a biological chamber containing secreted angiogenic and osteogenic factors around the bone defect. We, therefore, performed a descriptive mass spectrometric analysis to compare the secretome profiles of PMMA-IMs and metakaolin-IMs. We detected a total of 688 proteins in both groups (Figure 4A), 683 (99.3%) of which were not differentially secreted between PMMA-IMs and metakaolin-IMs (i.e., the frequency of secretion of these proteins was similar in the two groups). In contrast, the secretion frequency differed between the two groups for five proteins (0.72%): four were more frequently secreted by metakaolin-IMs, and one was more frequently secreted by PMMA-IMs. The four proteins more frequently secreted by metakaolin-IMs were identified as cysteine- and glycine-rich protein 3, the GON7 subunit of the KEOPS complex, carboxylic ester hydrolase and synaptogyrin. These proteins are involved in various metabolic pathways, including myogenesis and apoptosis. The protein most frequently secreted by PMMA-IMs was the neurotrophin tyrosine kinase receptor 1 TrkA L0 variant, which is involved in the MAPK pathway.

We also investigated the expression of the pro-osteogenic mediator BMP-2 within the membranes by immunohistochemistry (Figure 4B). BMP-2-expressing cells were uniformly distributed throughout the membranes, but BMP-2 staining was more intense in metakaolin-IMs than in PMMA-IMs. Furthermore, the percentage of the membrane area positive for BMP-2 was 1.9 times higher in metakaolin-IMs than in PMMA-IMs (25.25% ± 4.83% versus 48.41% ± 7.11%, *p* = 0.0.21).

### 3.3. Macrophage Distribution in IMs

We characterized the macrophage populations in IMs by immunofluorescence analysis to detect both CD68 and CD206, with CD68 used as a phenotypic marker of the M1-like subtype and CD68+/CD206+ double labeling as a marker of the M2-like subtype (Figure 5A). CD68-/CD206+ cells were defined as satellite cells. Semi-automatic quantification revealed that the total macrophage population was significantly larger in metakaolin-IMs than in PMMA-IMs (25.77% ± 5.48% versus 48.11% ± 5.77%, *p* = 0.02; Figure 5B). This larger total macrophage population reflected a significant expansion of the M1-like population (20.01% ± 3.81% versus 36.30% ± 4.45%, *p* = 0.02) and a smaller, non-significant expansion of the M2-like subtype (5.75% ± 2.40% versus 11.81% ± 1.72%, *p* = 0.07).

### 3.4. Bone Healing after IMT Stage 2 Surgery

We compared the bone-healing properties of PMMA-IMs and metakaolin-IMs, by performing a quantitative analysis of callus volume within the osteotomy region 10 weeks after bone graft implantation in the IM cavities. We observed a small, non-significant difference in new bone volume within the defect, with a slightly greater volume in the metakaolin group (16.65% ± 3.59% versus 26.37% ± 6.5%, *p* = 0.22, Figure 6A) than in the PMMA group.

## 4. Discussion

In this study, we evaluated the use of a metakaolin spacer as an alternative to standard PMMA spacers for the induced membrane technique. This technique is increasingly used in orthopedic surgery to repair large bone defects in humans. We first analyzed the osteogenic, biological and inflammatory properties of IMs in rat bone defects treated with metakaolin or PMMA spacers. We then assessed bone repair efficiency in rats 10 weeks after the implantation of a morselized corticocancellous allograft into the IM cavity generated by the two types of spacers.

### 4.1. Metakaolin Modifies Several Osteogenic and Biological Parameters of IMs

IMs are well-organized bilayer encapsulation membranes resulting from a foreign body reaction to the implanted spacer [15,24,25]. In a previous investigation, we demonstrated the importance of both the cellularity and collagen density of IMs on their biological properties in humans. Indeed, patients in which the Masquelet technique was unsuccessful (absence of bone repair resulting in non-union) had IMs with 50% lower levels of cellularity and a much higher collagen density (fibrosis-like status membrane) than those in which this technique was successful [4]. Conversely, here, the replacement of the PMMA spacer with a metakaolin spacer tended to increase IM cellularity (+30%). The IM acts as a biological chamber, promoting bone graft vascularity and corticalization by the secretion of various cytokines and growth factors [3,5,26,27,28]. Mass spectrometry showed that the metakaolin-IM and PMMA-IM secretomes differed by only 0.72%, suggesting that the secreted protein profiles of metakaolin-IMs and PMMA-IMs differed very little. Our mass spectrometry proteomic analysis was purely descriptive. We did not, therefore, have precise data for protein secretion levels. However, evidence from other molecular and protein analyses suggests that the expression levels of several proteins are modified by metakaolin-IMs. We observed a non-significant trend towards higher *Igf-1*, *Il-6* and *Il-1β* transcript levels with the metakaolin spacer. Fischer et al. [29] showed that serum *Igf-1* levels were higher in patients successfully treated with the IMT than in patients presenting treatment failure.

We found that *Tgf-β* transcript levels and BMP-2 protein levels were significantly higher (2.7-fold and 1.9-fold increases, respectively) in metakaolin-IMs than in PMMA-IMs. BMP-2 is undoubtedly the most osteoinductive growth factor, promoting the migration, proliferation and osteoblastic differentiation of osteoprogenitor cells. TGF-β has dual activity in bone remodeling activity [30], acting as a chemoattractant for osteoprogenitor cells at bone lesion sites and stimulating bone formation (osteoprogenitor proliferation and active osteoblastic differentiation; collagen synthesis) while inhibiting bone resorption (inhibition of osteoclast proliferation and activity). Interestingly, BMP-2 and TGF-β belong to the same growth factor superfamily. They bind to serine/tyrosine kinase receptors, and this interaction activates the SMAD intracellular signaling transduction pathway, which is involved in various steps of the bone regeneration process during fracture healing [31]. Tang et al. [32] suggested that activation of the SMAD pathway by both BMP-2 and TGF-β might underlie the osteogenic effects mediated by IMs. Taken together, the increase in IM cellularity and higher levels of IGF-1, IL-6, IL-1β IM transcripts and BMP-2 protein are consistent with the theory that the membranes induced by metakaolin are more osteogenic than those induced by PMMA spacers.

### 4.2. Metakaolin Spacers Modulate the IM Immune Microenvironment

The nature of the spacer did not affect systemic inflammation, as estimated from white blood cell counts in our animals. However, the increases in *Igf-1*, *Il-6*, *Il-1β* and *Tgf*-*β* transcripts suggested that the metakaolin spacer modulated the local inflammatory response. Macrophages are one of the most abundant sources of cytokines [33]. In this context of biomaterial implantation in a bone lesion area, it is difficult to separate the local inflammation process induced by the bone lesion from that triggered by biomaterial implantation. Macrophages form a highly heterogeneous and plastic population of cells and are, therefore, of particular interest in the IMT context due to their involvement in both wound repair processes and the foreign body response [34,35]. After activation, tissue-resident and monocyte-derived macrophages are recruited to the inflammation site. Depending on local environment cues, they transiently gain and lose functions by undergoing major phenotypic changes. A consensus has emerged concerning a sequential macrophage polarization pattern in the bone-healing process [35,36,37]. Following the formation of the bone lesion, there is a rapid, massive infiltration of monocytes and undifferentiated M0 subtype macrophages at the fracture site. During the first few days after the injury, the polarization of macrophages to the M1 “pro-inflammatory” phenotype is driven by secreted inflammatory and chemoattractant mediators, such as IL-6, IL-1β, IFN-γ, TNFα and monocyte chemotactic protein 1 or MCP-1.

M1 macrophages remove the provisional fibrin matrix and necrotic cells by phagocytosis. By secreting TNFα, IL-1β, IL-6 and MCP-1, they support inflammation by recruiting additional immune cells, but they also initiate the recruitment of fibroblasts and osteoprogenitor cells to the lesion site. Later in inflammation/repair kinetics, under the influence of IL-4, IL-10 and IL-13 signaling, macrophage polarization switches to the M2 “anti-inflammatory” phenotype. The secretion of VEGF, matrix metalloproteinases (MMPs), BMP-2 and platelet-derived growth factor (PDGF) by M2 macrophages triggers both angiogenesis and bone tissue remodeling during the healing process [35,37]. Macrophages are also crucial regulators of the FBR [38]. Following the implantation of biomaterials, plasma components adsorb onto the surface of the material, promoting neutrophil inflammation and macrophage recruitment. The macrophage-driven secretion of TGF-β around the implant triggers the transdifferentiation of fibroblasts into myofibroblasts, thereby promoting myofibroblast collagen production, leading to encapsulation of the biomaterial. In addition, macrophages fuse to form foreign body giant cells (FBGCs). FBGCs are large multinucleated cells secreting cathepsin-K and reactive oxygen species to degrade the foreign body (in this case, the biomaterial). 

Surprisingly, despite their key role in the FBR, little is known about the phenotypes of macrophages in vivo during this reaction. Conflicting reports have been published [39], probably because the characteristics of the biomaterial (including surface chemistry, porosity, stiffness, etc.) directly affect macrophage phenotype. However, there is a general consensus that both M1 and M2 macrophages are present throughout the FBR [40,41]. Moreover, higher levels of M2 macrophages than M1 macrophages surrounding implanted biomaterials are associated with more constructive remodeling [42,43]. For example, Zhu et al. [44] tested the capacity for orienting macrophage polarization of four scales of honeycomb-like titanium structures with honeycomb diameters ranging from 90 nm to 5 µm. Raw 264.7 macrophages cultured with the smoothest titanium structure had the highest M2-macrophage polarization rate, with the highest levels of CD206 expression (a specific marker of M2 macrophages) and IL-4, IL-10 and BMP-2. In vivo, the implantation of titanium rods with 90 nm honeycombs in rat tibia gave the best results for bone osteointegration [44]. Here, we compared macrophage polarization in IMs according to the nature of the spacer implant. In metakaolin-IMs, we observed a significant expansion of the total macrophage population. This observation is consistent with previous findings indicating that aluminosilicates stimulate the immune response by inducing the activation of macrophages. The expansion of the macrophage population in metakaolin-IMs is also consistent with the increase in key inflammation-related transcripts observed in the same membranes. More specifically, even though the expansion of the M2-like population was not significant (*p* = 0.07), both M1-like and M2-like macrophage subtypes increased markedly in frequency in metakaolin-IMs. To our knowledge, this study is the first in an IMT context to show a link between greater osteogenic properties of the induced membrane and a spacer-driven modulation of the phenotype and number of macrophages. Further studies are required to elucidate the mechanism underlying the balance between M1 and M2 macrophages in the induced membrane. 

### 4.3. Metakaolin Slightly Improves Bone Repair Efficiency

Given the more osteogenic properties of the metakaolin-IMs, better bone regeneration was expected in this group. Unsurprisingly, we observed a trend towards better bone healing, as shown by the 1.58-fold increase in BV/TV when a metakaolin spacer was used to generate IM rather than a PMMA spacer. Other alternative biomaterials have been tested for the creation of IM mimetics [45] or the induction of IMs with enhanced osteogenic properties [46]. However, mixed results for bone repair outcomes have been obtained for these alternative spacers. Indeed, after four weeks of maturation in rats, smooth and rough titanium spacers generated thicker IMs than smooth and rough PMMA spacers but with similar histological structures and biochemical expression parameters [47]. The only difference observed concerned IL-6 protein levels in the IM, which were about 35% higher with rough spacers (both PMMA and titanium spacers) than with smooth spacers. Smooth PMMA spacers resulted in a more functional bone union than the other PMMA and titanium spacers tested [47]. Unfortunately, the authors did not investigate the immune microenvironment of the membranes, particularly the balance between M1 and M2 macrophages. Following on from the successful clinical use of polypropylene syringes as alternative spacers to PMMA cement to treat metacarpal bone lesions [48], we validated the potential of this biomaterial in a rat IMT model [15]. Polypropylene-induced membranes had a similar histologic organization, cell density and BMP-2 protein level to PMMA- IMs, and similar levels of serum bone turnover markers. In micro-CT analysis, bone regeneration capacities were similar in the polypropylene and PMMA groups of rats [15]. Our investigation highlights the value of polypropylene syringes as an alternative to PMMA cement for use as spacers in a military practice context and/or in low-medical resource environments. With a view to developing a modified IMT approach for efficient one-step surgery, Ma et al. [49] evaluated the osteogenic properties of calcium sulfate (CS)-induced membranes in rats. The histological characteristics of CS-IM and PMMA-IM were similar, except that the calcium sulfate spacer induced thicker membranes. Levels of the TGF-β1, BMP-2 and VEGF proteins were not significantly higher in CS-IMs at two, four, six and eight weeks post-implantation, whereas IL-6 protein levels were significantly higher in PMMA-IMs at two weeks post-implantation. Finally, CS-IMs promoted better endochondral ossification at the edges of the bone defect than PMMA-IMs at six and eight weeks post-implantation [49]. The authors concluded that calcium sulfate could replace PMMA as an alternative spacer in IMT. The results obtained with the metakaolin spacer in this study are equivalent to those obtained by Ma’s research team for the calcium sulfate spacer.

## 5. Limitations, Conclusions and Future Directions

This study is the first to date to investigate a correlation between local inflammation/the immune microenvironment of the IM and osteogenic properties by comparing IMs generated with PMMA and metakaolin spacers in a preclinical rat model of IMT. Metakaolin induced a membrane with slightly better osteogenic properties than the PMMA spacer, improving bone-healing efficiency, albeit not significantly in our rat model. This significant success in bone repair was accompanied by an expansion of the macrophage population in the IM structure for both M1 and M2 subtype macrophages. This stronger local inflammation process was sustained by local overexpression of the osteogenic BMP-2 protein and several inflammatory cytokines, including TGF-β, IL-1β and IL-6.

This study had several limitations. The number of rats included was relatively small, which may have contributed to the high standard deviation in the RT-PCR analysis. Furthermore, our study included only male rats. Sex-specific differences in bone-healing outcomes remain underinvestigated, but most studies in the field have suggested that being female is a significant risk factor for compromised bone healing [50]. This influence of sex on fracture healing may be related to the smaller numbers of mesenchymal stromal cells (MSCs) in female bone marrow [51].

Given the presence of MSCs in the induced membrane [4], it would be desirable to investigate sex-specific differences in bone-healing outcomes, particularly in the IMT context. Another limitation of the study concerns the in situ characterization of polarized macrophages using CD68 and CD206 immunofluorescence. The CD68 protein is one of the most common monocyte/macrophage markers [52], whereas the CD206 protein is mostly expressed by M2 macrophages [53]. The co-expression of CD68 and CD206 is generally considered to indicate an M2-like phenotype. In this study, the expression of CD68 alone was considered to indicate a M1-like macrophage phenotype. The use of a single marker for identifying the M1-like population is questionable. Since a weak CD68 expression can be detected in some non-hematopoietic cells (mesenchymal stem cells, fibroblast, endothelial and tumor cells) [54], we assumed that the M1-like population is overestimated in our study. Indeed, the in situ detection of polarized macrophages is technically challenging [22,55], and none of the other discriminating markers we tested gave conclusive results. We acknowledged that the in situ CD68-based strategy for M1-like cell detection can be regarded as a “by default” identification of this population. Although there is no real direct evidence for the use of CD68 as a single marker for the M1-type population characterization in rats, this labeling approach is commonly described in the literature [22,56,57,58], thus providing robust indirect evidence to our conclusion.

In conclusion, metakaolin spacers would be a valuable biomaterial for replacing PMMA spacers in the Masquelet technique. One particularly interesting clinical application would be the healing of complicated bone defects. Indeed, this strategy would involve the manufacture of 3D printing molds in the shape of the injured bone areas based on the CT scans for the patients concerned. A metakaolin spacer could then be molded in a specific cast to obtain the appropriate shape before implantation into the bone defect. Finally, given the high absorbency of metakaolin, the metakaolin spacer could be impregnated with a large panel of antibiotics to eradicate potential bone infections that might lead to a failure of bone repair. Conversely, only heat-resistant antibiotics could be loaded onto PMMA spacers due to the exothermic nature of the PMMA polymerization reaction.

## Figures and Tables

**Figure 1 biomedicines-10-03017-f001:**
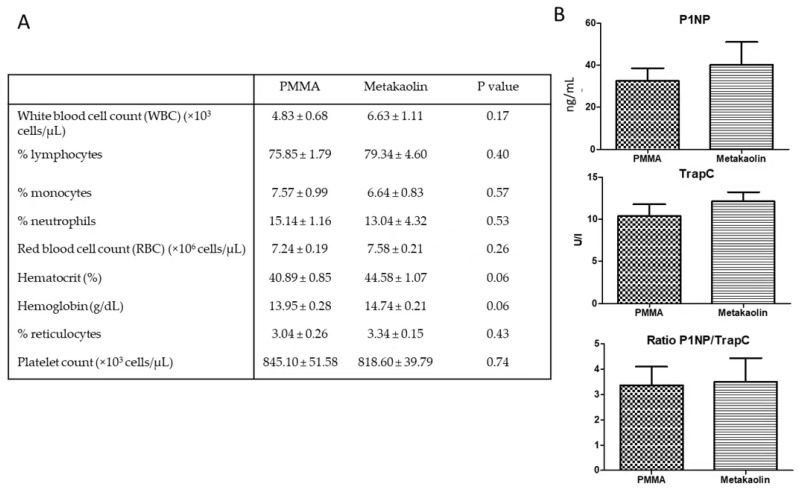
Panel (**A**) shows the hematological parameters of animals four weeks after spacer implantation. Blood was collected into EDTA-containing tubes when the animals were killed, and blood parameters were determined with an optical hematology analyzer (MS-9, Melet Schloesing) with rat-specific analysis software. Panel (**B**) shows the serum levels of bone turnover markers, as determined by ELISA, four weeks post-spacer implantation. Concentrations of a bone formation marker P1NP (**top**) and a bone resorption marker TRAP-C (**middle**) were determined, and turnover for bone remodeling was evaluated by calculating the P1NP/TRAP-C ratio (**bottom**).

**Figure 2 biomedicines-10-03017-f002:**
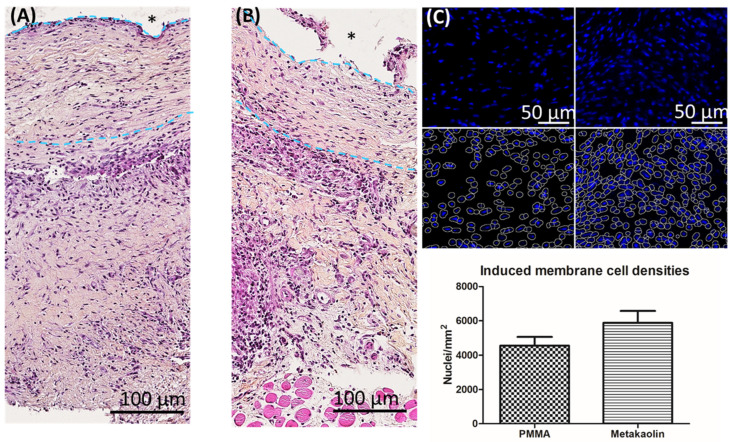
Representative hematoxylin-eosin-saffron-stained sections of (**A**) PMMA-IMs and (**B**) metakaolin-IMs showing their histological organization. * Indicates the site of the spacer before its removal. The right panel (**C**) illustrates the semi-automatic counting process of DAPI-stained nuclei and the comparison of cell density (the mean number of nuclei per mm^2^ ± SEM) in PMMA-IMs and metakaolin-IMs.

**Figure 3 biomedicines-10-03017-f003:**
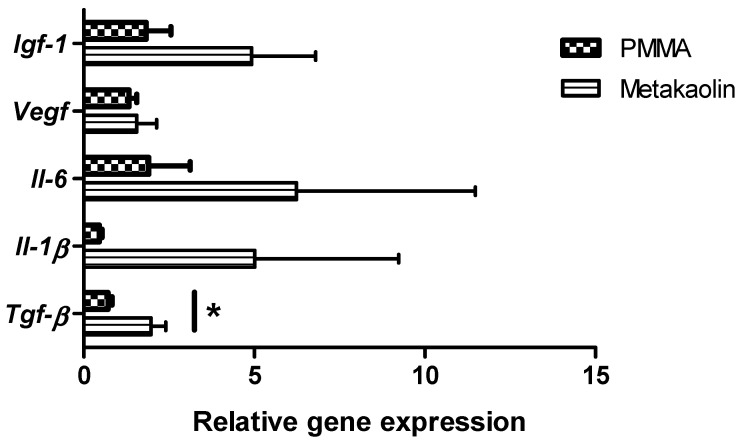
Relative levels of *Igf-1*, *Vegf*, *Il-6*, *Il-1β* and *Tgf-β* mRNA in four-week-old PMMA-IMs or metakaolin-IMs. Data are expressed as the mean ± SEM, * *p* < 0.05.

**Figure 4 biomedicines-10-03017-f004:**
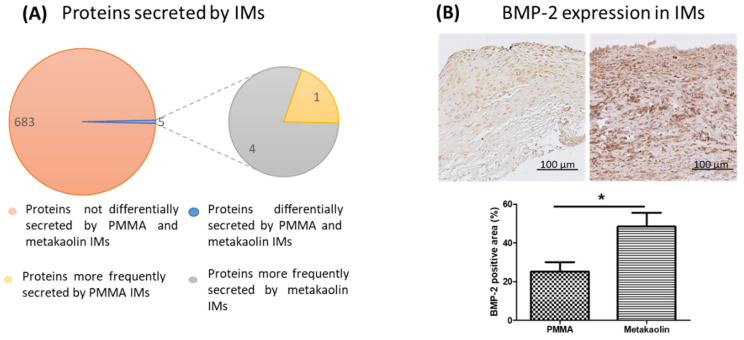
(**A**) Secretome profiles of PMMA-IMs and metakaolin-IMs. In total, 683 proteins with similar frequencies of secretion in the PMMA and metakaolin groups were identified by mass spectrometry. By contrast, five proteins were differentially secreted: four proteins were more frequently secreted by the metakaolin-IMs, and the other was more frequently secreted by PMMA-IMs. (**B**) Representative histological slide of in situ BMP-2 immunostaining in PMMA-IMs (left panel) and metakaolin-IMs (right panel). The diagram shows the percentage of the area of the collected membranes positive for BMP-2; * *p* < 0.05.

**Figure 5 biomedicines-10-03017-f005:**
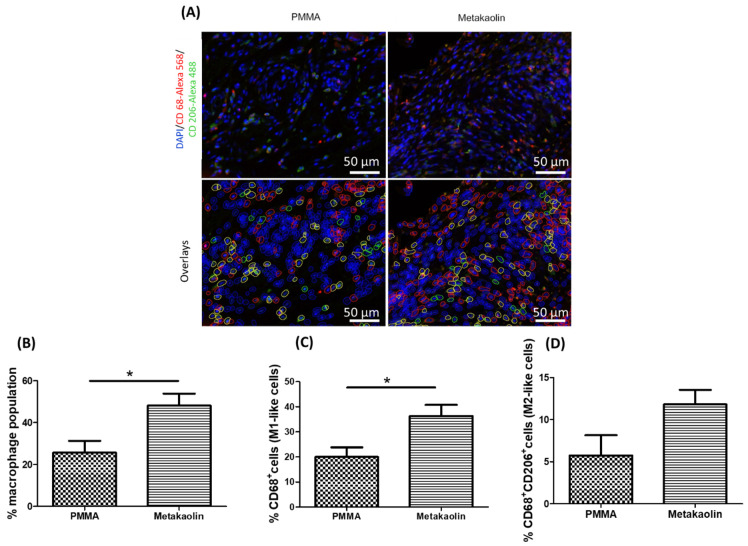
Identification and quantification of M1-like and M2-like macrophages in PMMA-IMs and metakaolin-IMs: (**A**, **top panel**) Representative immunolabeling with anti-CD68 (red) and anti-CD206 (green) antibodies and DAPI (blue) nuclear staining. (**A**, **bottom panel**) Illustration of semi-automatic macrophage quantification. Green objects represent satellite cells, and red and yellow objects correspond to M1-like and M2-like macrophages, respectively. Histograms show (**B**) the % total (M1-like + M2-like) macrophages in IMs, (**C**) the % M1-like macrophages and (**D**) M2-like macrophages. * *p* < 0.05.

**Figure 6 biomedicines-10-03017-f006:**
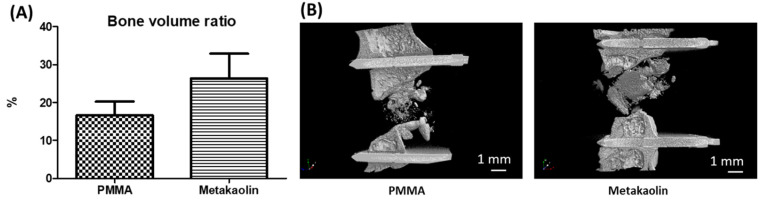
Bone formation was assessed by microCT. (**A**) The quantitative and comparative analysis showed no difference in bone volume between the metakaolin and PMMA groups. (**B**) Representative three-dimensional reconstructions of the region of interest in the two groups.

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
