# Peer review of "Influence of the Immune Microenvironment Provided by Implanted Biomaterials on the Biological Properties of Masquelet-Induced Membranes in Rats: Metakaolin as an Alternative Spacer"

_biomedicines, 2022, doi:10.3390/biomedicines10123017_

Round 1

Reviewer 1 Report

Overall suggestion: Accept with minor changes

Minor suggestions:

1.     Please break down the 1st paragraph of the introduction into 2-3 paragraphs to make it easier for the readers. Currently it appears as a bulk of information with no segregation of the sequence in which you introduce the relevant topics to us.

2.     While looking online I found these two very recent references which I think are relevant to your work. Please do consider including them in your introduction: a - https://doi.org/10.1016/j.injury.2021.11.003 and b - https://doi.org/10.1016/j.injury.2022.02.036

3.     For section 2.1 – please mention the gender of the rats used and the justification if it includes only one gender. If you have used rats from one gender only – then please acknowledge this in your discussion and also outline it as a limitation as a critique to your work.

4.     For figure 3 – please add the units in which the gene expression has been measured (Relative expression) not just in the figure legend but also in the x-axis of the figure.

5.     For gene expression – please mention the gene to which all the data is relative to, i.e. please mention the housekeeping gene and how exactly you calculated the values plotted on the graph in figure 3.

6.     Figure 5a can be rather subjective, especially if analysed by one person only. Ideally, it would be nice to have the same data analysed in a blind fashion by at least two of the authors/ scientists and present the data as an average to reduce potential bias when analysed by one person only.

7.     Kindly rephrase section 5 as ‘Limitations, conclusions and future directions’ add a paragraph each on limitation as well as on your future vision of the current work. Limitation must reflect on your work and acknowledge the gaps in your current study that could be improved and future directions must clearly indicate in which direction you see your current work going towards for clinical applications in humans.

Author Response

First, we would like to thank the reviewers for their interest in our work and for their fair comments on our manuscript. Answers to the reviewer’s comments are detailed below. Exept for the english proofreading, all revisions are indicated in red characters in the manuscript text.

Besides, we would like to mention that we made some changes regarding affiliations of two authors (FP and DL).

Open Review

( )I would not like to sign my review report
(x)I would like to sign my review report

English language and style

( )Extensive editing of English language and style required
( )Moderate English changes required
(x)English language and style are fine/minor spell check required
( )I don't feel qualified to judge about the English language and style

As suggested, the English grammar/style of the manuscript has been corrected by a native english speaker (alexedelman.com). We hope that style errors have been correctly addressed.

Yes

Can be improved

Must be improved

Not applicable

Does the introduction provide sufficient background and include all relevant references?

( )

(x)

( )

( )

Are all the cited references relevant to the research?

(x)

( )

( )

( )

Is the research design appropriate?

(x)

( )

( )

( )

Are the methods adequately described?

( )

(x)

( )

( )

Are the results clearly presented?

( )

(x)

( )

( )

Are the conclusions supported by the results?

(x)

( )

( )

( )

Comments and Suggestions for Authors

Overall suggestion: Accept with minor changes

Minor suggestions:

  1. Please break down the 1st paragraph of the introduction into 2-3 paragraphs to make it easier for the readers. Currently it appears as a bulk of information with no segregation of the sequence in which you introduce the relevant topics to us.

As requested, we broke down the 1st paragraph into 3 paragraphs. In addition, we cut down some « non-essential » informations (including the FBR process sequences) to lighten the text and make it easier to the reader.

  1. While looking online I found these two very recent references which I think are relevant to your work. Please do consider including them in your introduction:

a -https://doi.org/10.1016/j.injury.2021.11.003

This citation was added in the introduction (see line 96)

b-https://doi.org/10.1016/j.injury.2022.02.036

This citation was added in the discussion (line 547), since it is more relevant than in the introduction to our opinion.

  1. For section 2.1 – please mention the gender of the rats used and the justification if it includes only one gender. If you have used rats from one gender only – then please acknowledge this in your discussion and also outline it as a limitation as a critique to your work.

Only male rats were used in the study. This information was included in line 131 (materials and methods). In many studies, data generated from female and male animals are pooled together, simply stipulating that there would be no sex-specific influence in the studied process. However, female sex was identified as a major risk factor for compromised bone healing by both clinical and experimental investigations. For this reason, we always exclusively include male rats in our studies. This point is now acknowledged in the discussion.

  1. For figure 3 – please add the units in which the gene expression has been measured (Relative expression) not just in the figure legend but also in the x-axis of the figure.

The unit (relative gene expression) was added to the x-axis.

  1. For gene expression – please mention the gene to which all the data is relative to, i.e. please mention the housekeeping gene and how exactly you calculated the values plotted on the graph in figure 3.

The 3 reference genes we used were already mentioned in the « materials and methods » section of the manuscript. Lines 225 and 226, we added « 3 references genes … » and « 5 target genes … » to lead readers to information.

Regarding calculation methods, the values were obtained by using the comparative threshold method [1]. In addition, real time PCR study was performed in keeping with the MIQE guidelines [2]. Both references were added in the text (lines 223 and 213).

  1. Livak, K.J.; Schmittgen, T.D. Analysis of Relative Gene Expression Data Using Real-Time Quantitative PCR and the 2(-Delta Delta C(T)) Method. Methods 2001, 25, 402–408, doi:10.1006/meth.2001.1262.
  2. Bustin, S.A.; Benes, V.; Garson, J.A.; Hellemans, J.; Huggett, J.; Kubista, M.; Mueller, R.; Nolan, T.; Pfaffl, M.W.; Shipley, G.L.; et al. The MIQE Guidelines: Minimum Information for Publication of Quantitative Real-Time PCR Experiments. Clin Chem 2009, 55, 611–622, doi:10.1373/clinchem.2008.112797.

  1. Figure 5a can be rather subjective, especially if analysed by one person only. Ideally, it would be nice to have the same data analysed in a blind fashion by at least two of the authors/ scientists and present the data as an average to reduce potential bias when analysed by one person only.

We understand reviewer’s remark about the potential subjectivity in the macrophage quantification analysis. When initiating this quantification, we ourselves were aware of inducing a potential experimenter bias in such an analysis. This is why we developed a semi-automatic program to avoid subjectivity. All slides were scanned following immunofluorescence stainings. Then, MD drew the regions of interest (ROIs) on the digitized version of the slides. By using Fiji software, MO created a program to automatically detect immunofluorescent labelings (Alexa Fluor 488 , Alexa Fluor 568  and DAPI) and then count cells into the ROIs. An upstream work was performed by MD and KN (researcher specialized in immunonofluorescence stainings) to validate the program.

  1. Kindly rephrase section 5 as ‘Limitations, conclusions and future directions’ add a paragraph each on limitation as well as on your future vision of the current work. Limitation must reflect on your work and acknowledge the gaps in your current study that could be improved and future directions must clearly indicate in which direction you see your current work going towards for clinical applications in humans.

As requested, we rephrased section 5 and made some limitations to the study regarding the small number of rats used in the study and the exclusion of female rats. We also discuss technical limitations on CD68 and CD206 antibodies used to phenotype macrophages in situ.

Reviewer 2 Report

This paper compared the immunological differences of standard PMMA and metakaolin after 4 weeks of implantation. The author used various methods, including histological, immunohistochemical, mass spectroscopy and gene expression to investigate the immunological differences. The author also found that metakaolin induced membranes improved bone regeneration by improving callus properties. I do have some comments regarding this paper.

1.       Do you have some material characterization results?

2.       CD68 is a macrophage pan-marker instead of M1 marker.

3.       Figure2. Please explain on what basis you conclude “an inner layer at the interface with the biomaterial including fibroblasts, lymphocytes and macrophages.”

4.       The author stated “Quantification of DAPI-stained-nuclei demonstrated that cellular xxxx”. Where is your DAPI stained nuclei?

5.       Figure3. Why is your standard deviation is so high?

6.       Figure4, why is your BMP2 staining is everywhere? Additionally, the nuclei staining is very weak.

7.       Figure5. Again, CD68 is macrophage pan-marker, therefore, your CD206 positive cells should also be CD68 positive cells. https://www.sciencedirect.com/topics/neuroscience/cd68

Round 2

Reviewer 2 Report

The author mentioned " Of note, CD68+/ CD206- labeling to identify M1-polarized macrophages on tissue sections by default can be also found in literature [9,10].” These two papers cited by the author are all human samples, however, the author is using rat animal models. The author should realize that rats and human are two different species. Sometimes a marker specifically expressed on human samples does not mean it will express on rodent species. Therefore, please provide direct evidence from previously published literature indicating CD68+/CD206- to identify M1-polarized macrophages, especially in rat samples.

Author Response

The author mentioned " Of note, CD68+/ CD206- labeling to identify M1-polarized macrophages on tissue sections by default can be also found in literature [9,10].” These two papers cited by the author are all human samples, however, the author is using rat animal models. The author should realize that rats and human are two different species. Sometimes a marker specifically expressed on human samples does not mean it will express on rodent species. Therefore, please provide direct evidence from previously published literature indicating CD68+/CD206- to identify M1-polarized macrophages, especially in rat samples.

We understand reviewer’s concern. In our recently published review on macrophage identification in situ,  we briefly mentioned that some markers specifically expressed on murine samples, for example F4/80, are not detected in human macrophages. Therefore, we are aware that the antibody’s choice for macrophage immunolabeling is species-dependent. This is another limitation of the technique.

Regarding literature, our team already published in Biomedicines a study using the single CD68 immunolabeling for M1 detection in rat tissues [1]. This reference is already cited in the manuscript. Several publications from other research teams using the same strategy in rat samples can be found in the literature [2–4]. All these references suggest that the use of CD68 labeling to target the M1 type population is still questionable, but sufficiently accepted to be published without risk of criticism.

  1. Nikovics, K.; Durand, M.; Castellarin, C.; Burger, J.; Sicherre, E.; Collombet, J.-M.; Oger, M.; Holy, X.; Favier, A.-L. Macrophages Characterization in an Injured Bone Tissue. Biomedicines 2022, 10, 1385, doi:10.3390/biomedicines10061385.
  2. Nakagawa, M.; Karim, M.R.; Izawa, T.; Kuwamura, M.; Yamate, J. Immunophenotypical Characterization of M1/M2 Macrophages and Lymphocytes in Cisplatin-Induced Rat Progressive Renal Fibrosis. Cells 2021, 10, 257, doi:10.3390/cells10020257.
  3. Tsuji, Y.; Kuramochi, M.; Golbar, H.M.; Izawa, T.; Kuwamura, M.; Yamate, J. Acetaminophen-Induced Rat Hepatotoxicity Based on M1/M2-Macrophage Polarization, in Possible Relation to Damage-Associated Molecular Patterns and Autophagy. Int J Mol Sci 2020, 21, E8998, doi:10.3390/ijms21238998.
  4. Hashimoto, A.; Karim, M.R.; Kuramochi, M.; Izawa, T.; Kuwamura, M.; Yamate, J. Characterization of Macrophages and Myofibroblasts Appearing in Dibutyltin Dichloride-Induced Rat Pancreatic Fibrosis. Toxicol Pathol 2020, 48, 509–523, doi:10.1177/0192623319893310.

In order to fully answer reviewer’s concern, references related to human samples in the manuscript were changed to the above-mentioned ones on rat samples.